# Biodiversity Impacts of Increased Ethanol Production in Brazil

**A.S. Duden** [1],*[ID]**, P.A. Verweij** [1][ID]**, A.P.C. Faaij** [2]**, D. Baisero** [3,4]**, C. Rondinini** [3]
**and F. van der Hilst** [1][ID]

[1] Copernicus Institute of Sustainable Development, Group Energy & Resources, Utrecht University, Princetonlaan 8a, 3584 CB Utrecht, The Netherlands; P.A.Verweij@uu.nl (P.A.V.); F.vanderHilst@uu.nl (F.v.d.H.)

[2] Center for Energy and Environmental Sciences, Faculty of Science and Engineering, University of Groningen, Nijenborgh 6, P.O. Box 221, 9700 AE Groningen, The Netherlands; A.P.C.Faaij@rug.nl

[3] Global Mammal Assessment Program, Department of Biology and Biotechnologies, Sapienza Università di Roma, Viale dell'Università 32, 00185 Roma, Italy; daniele.baisero@gmail.com (D.B.); carlo.rondinini@uniroma1.it (C.R.)

[4] Key Biodiversity Area Secretariat and Wildlife Conservation Society, c/o RSPB, David Attenborough Building, Pembroke Street, Cambridge CB2 3QZ, UK

[*] Correspondence: A.S.Duden@uu.nl; Tel.: +31-(0)30-253-6967

**Abstract:** Growing domestic and international ethanol demand is expected to result in increased sugarcane cultivation in Brazil. Sugarcane expansion currently results in land-use changes mainly in the Cerrado and Atlantic Forest biomes, two severely threatened biodiversity hotspots. This study quantifies potential biodiversity impacts of increased ethanol demand in Brazil in a spatially explicit manner. We project changes in potential total, threatened, endemic, and range-restricted mammals' species richness up to 2030. Decreased potential species richness due to increased ethanol demand in 2030 was projected for about 19,000 km$^2$ in the Cerrado, 17,000 km$^2$ in the Atlantic Forest, and 7000 km$^2$ in the Pantanal. In the Cerrado and Atlantic Forest, the biodiversity impacts of sugarcane expansion were mainly due to direct land-use change; in the Pantanal, they were largely due to indirect land-use change. The biodiversity impact of increased ethanol demand was projected to be smaller than the impact of other drivers of land-use change. This study provides a first indication of biodiversity impacts related to increased ethanol production in Brazil, which is useful for policy makers and ethanol producers aiming to mitigate impacts. Future research should assess the impact of potential mitigation options, such as nature protection, agroforestry, or agricultural intensification.

**Keywords:** species richness; bioenergy; sugar cane; land-use modeling; mammals; biodiversity; bioethanol; land-use change; habitat modeling; biofuel

## 1. Introduction

Brazil is one of the major producers and exporters of agricultural and forestry products in the world [1,2]. Crop production in Brazil is dominated by the production of sugarcane (30% of net production value), soybean (29%), and corn (4%) [2], and the area dedicated to the cultivation of these crops has increased rapidly over the last 30 years [2]. Brazil currently has the largest area of sugarcane cultivation in the world [2], and is the second largest producer of ethanol [3]. The historical increase in sugarcane cultivation is primarily the result of Brazilian policies focused on stimulating the production of sugarcane-based ethanol in order to increase energy security, promote rural development, and decrease the dependency on fossil fuels [4]. Due to growing domestic and international demand, Brazilian ethanol production is expected to increase from 33.3 billion liters in 2018/2019 [5] up to

54.2 billion liters in 2030 [6] based on the global outlook of International Energy Agency (IEA) and Organization for Economic Cooperation and Development (OECD) [7].

Today, sugarcane is grown mainly in the Cerrado and Atlantic Forest biome in the southeast of Brazil, and has recently also expanded further into the northwestern part of the Cerrado biome [4,8]. The projected increase in ethanol production is expected to require additional land for sugarcane cultivation, resulting in changes in land use, which are both direct, when sugarcane replaces an area previously occupied by another land use, or indirect, when the expansion of sugarcane induces changes in land use elsewhere [6,9–11]. About 35,000 km$^2$ of sugarcane expansion is projected to occur between 2012 and 2030, mostly in the Cerrado and Atlantic Forest biomes [6,11]. Sugarcane is expected to replace mainly cropland and natural grassland [6,10], as well as shrubland [11]. Additionally, the projected expansion of sugarcane is expected to result in indirect land-use changes affecting an area ranging from about 20,000 [6,11] to 78,000 km$^2$ [10]. The three biomes where direct and indirect land-use change related to sugarcane expansion were projected were the Cerrado, Amazon, and Atlantic Forest biomes [6,10,11], resulting in loss of forest [6,10] as well as other natural vegetation [6,11]. The Cerrado and Atlantic Forest biomes are both biodiversity hotspots [12] under severe threat [13,14].

Brazil is recognized as one of the most biodiverse countries in the world [15,16] and contains about 15% to 20% of all species worldwide [17]. Mammal species are vulnerable to habitat loss, and Brazil has been identified as one of the top 10 of countries with a high projected mammal decline by 2050 [17,18]. The tropical and semi-deciduous Atlantic Forest is home to a large number of endemic species [15]. Currently, only about 12% of the original vegetation cover of Atlantic Forest is left, and the remaining forest areas are highly fragmented [19,20]. The Cerrado biome consists of savannah vegetation and contains a large share of endemic plant and vertebrate species [21,22]. Only about 2% of the Cerrado biome is legally protected [23,24], and over 40% of its original extent has been converted to agricultural use [13,23,25]. Almost 70% of the Amazon rainforest falls within the borders of Brazil [26], and the Brazilian Amazon forest has lost 18% of its original extent and has recently shown the highest levels of deforestation in a decade [27]. For a quarter of all Amazonian mammal species, further deforestation of the Amazon is projected to result in the loss of over 40% of their range by 2050 [28].

Past expansion and intensification of agriculture have been identified as the main drivers of species decline, and with ongoing agricultural expansion, biodiversity impacts are expected to become exacerbated in the future [29]. Conversion of native vegetation to agriculture is one of the main drivers of habitat loss in the tropics [30,31]. Expansion of sugarcane leads directly or indirectly to habitat loss [6], and has therefore been linked to impacts on biodiversity [32]. A number of studies have projected substantial future biodiversity impacts of agricultural expansion in Brazil. For example, studies assessing the projected effects of different scenarios of agricultural expansion pinpoint the Cerrado and Atlantic Forest as regions with large potential range contractions for a number of bird species [33] and non-flying mammals [34] by 2050, and potentially resulting in local extinction of 140 to 191 bird and mammal species [35] as well as 480 endemic plant species [21]. However, it remains unclear which proportion of these impacts can be attributed specifically to an increase in ethanol demand and subsequent sugarcane expansion, and how these biodiversity impacts vary over different regions in Brazil. Despite the strong projected increase in sugarcane production, the historical impact of agricultural expansion in Brazil, and the importance of Brazil in global biodiversity conservation, it has been noted that there is a paucity of studies that assess the potential impact of sugarcane expansion on biodiversity in Brazil [8]. The impact of ethanol-driven sugarcane expansion on biodiversity has therefore been identified as a priority for future research [4].

Biodiversity within a land-use type was found to vary spatially due to spatial heterogeneity in biophysical conditions [36], which implies that a conversion of one land-use type to another may result in different impacts on biodiversity, depending on the location. Therefore, it is important to assess potential biodiversity impacts in a spatially explicit manner. The aim of this study was to assess the potential impacts of ethanol-driven sugarcane expansion on biodiversity in Brazil. To this end, we performed a spatially explicit assessment of changes in potential mammal species richness following

scenarios of land-use change up to 2030. We focused on total, threatened, endemic, and range-restricted mammal species, and included the whole of Brazil in order to assess the biodiversity impacts of both direct and indirect land-use changes related to increased ethanol demand.

## 2. Materials and Methods

### 2.1. General Approach

This study combined land-use change projections for Brazil from previous studies [6,11] with species distributions derived from habitat suitability data [37] in order to assess impacts of ethanol-driven land-use change in Brazil on potential mammal species richness.

A spatial modelling approach developed in an earlier study [36] was applied to determine potential mammal species richness in 2012 and 2030 for a reference and an ethanol scenario. In the ethanol scenario, demand for ethanol was assumed to increase to meet global ethanol mandates from $24 \times 10^9$ L in 2012 to $54 \times 10^9$ L in 2030, while in the reference scenario, ethanol demand remained static at the 2013 level. Both scenarios also included changes in demand for other agricultural and silvicultural commodities. In this analysis, we combined land-use projections (at the 5-km resolution) and species habitat suitability maps (at the 300-m resolution) to create maps of projected species richness index (at the 5-km resolution) (Figure 1). Habitat suitability maps for all individual Brazilian mammal species were summed to produce a map of potential mammal species richness in Brazil (Figure 1).

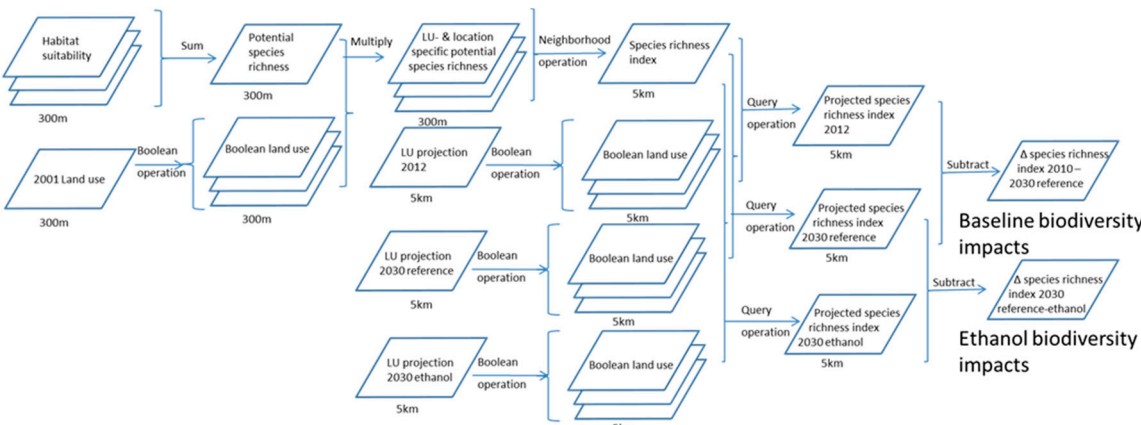

**Figure 1.** Overview of the approach applied in this study. Rectangles represent maps, stacked rectangles represent sets of maps. Text inside the rectangles describes the type of map, text above and below the arrow describes the analysis step. Text below the rectangles provides the resolution of the map.

We determined potential species richness for total, threatened, endemic, and range-restricted mammal species. Potential species richness in 2012 was compared to potential species richness in 2030 in the reference scenario to determine biodiversity impacts due to increased demand for agricultural and silvicultural commodities other than ethanol. Potential species richness in 2030 in the ethanol scenario was compared to potential species richness in 2030 for the reference scenario in order to quantify the impact of increased ethanol demand on biodiversity.

### 2.2. Biodiversity Indicators

Species richness is a frequently used indicator for biodiversity status [12,38,39]. In this study, we assessed biodiversity impact expressed as changes in potential species richness of mammals. We focused on the species richness of mammals because habitat loss and degradation have impacted mammal communities in such a way that globally, 25% of all mammal species are currently threatened with extinction [18,40,41]. Mammals are also one of the most intensively studied taxa [37], for which spatial habitat data is available globally at a high resolution [37]. Furthermore, mammal species

can modify vegetation structure and thereby influence species diversity and composition in other taxonomic groups [42]. Land-use changes may affect different mammal species in different ways. In this study, we used the following potential mammal species richness indicators: Total species richness, threatened species richness, endemic species richness, and restricted-range species richness. We used potential threatened, endemic, and range-restricted species richness as indicators because these species are considered conservation priorities due to vulnerability and high extinction risk. Threatened species are considered to be already at risk of extinction, and are therefore considered a conservation priority [43]. Endemic species are those species whose distribution is restricted to a unique geographical area [44] and are considered conservation priorities because of their uniqueness and high risk of extinction [44,45]. Species with restricted ranges are considered a conservation priority because they are more vulnerable to human impact in comparison to species with large ranges [46].

We selected all mammal species whose potential habitat overlapped partly or entirely with Brazil, which was the case for 610 species (Table 1; for a list of species, see the Supplementary Materials). Threatened mammal species were defined as 'critically endangered', 'endangered', or 'vulnerable' according to the International Union for Conservation of Nature (IUCN) Red List of Threatened Species [40] or the Brazilian National List of Threatened Fauna Species (Lista Nacional Ofical de Espécies da Fauna Ameaçadas de Extinção [47]). Endemic species were defined as having 100% of the potential habitat within Brazil, in line with previous research [15]. Restricted range species were identified as those species with a total potential habitat area smaller than 50,000 km$^2$, in accordance with earlier studies [48,49]. Based on our definition, there can be an overlap in the number of threatened, endemic, and range-restricted species. This means that a certain species can, for instance, be categorized as both threatened and range restricted.

**Table 1.** Numbers of mammal species included in this study; total number of species, threatened species, endemic species, and range-restricted species in Brazil, as derived from habitat suitability data [37].

| Species Group Indicator | Number of Species |
| --- | --- |
| All mammal species | 610 |
| Threatened mammal species | 107 |
| Endemic mammal species | 93 |
| Restricted-range mammal species | 120 |

### 2.3. Land-Use Projections

The projected land-use changes in Brazil following scenarios of future ethanol demand and developments in demand for other agricultural an silvicultural commodities were taken from previous studies [6,11]. We compared two scenarios: An ethanol scenario and a hypothetical reference scenario. The ethanol scenario assumed fulfillment of current and planned ethanol mandates worldwide, which results in Brazilian ethanol production more than doubling from $24 \times 10^9$ L in 2012 to $54 \times 10^9$ L in 2030 [11]. The reference scenario assumed static ethanol demand at the level of 2013 ($27 \times 10^9$ L) up to 2030, but in this scenario, the area of sugarcane still increases due to increased demand for sugar [11]. Both scenarios also assumed increased production of crops, livestock, and wood products by 2030 [6]. Trends of agricultural and silvicultural production were based on the Shared Socio-economic reference Pathway SSP2 (a 'Middle of the Road' scenario), which projects socio-economic developments up to 2100 in line with historical trends [50].

The land-use projections were based on a combination of the global computable general equilibrium (CGE) model MAGNET (Modular Applied GeNeral Equilibrium Tool) with the PLUC (PC Raster Land Use Change) model, which were run for the period 2012 to 2030. MAGNET was used to model the development in the demand and supply of different commodities, including ethanol, on a global scale [51]. It provided projections of the amount of land in Brazil needed to meet increased demand for ethanol up to 2030, as well as that of other agricultural and silvicultural commodities [6,11].

Projected land requirements between 2012 and 2030 for crops, livestock production, forest plantations, and sugarcane were then spatially allocated using the PLUC model [52,53]. PLUC was developed to project land-use change based on temporal developments in demand for various land uses and spatially explicit suitability of land for various land uses [52]. PLUC produced annual maps of projected land use in Brazil between 2012 and 2030 at a 25-km$^2$ resolution for the following land-use types: Grass and shrubs, natural forest, planted forest, crops (excluding sugarcane), sugar cane, planted pasture, rangeland, abandoned agricultural land, urban, water, and bare soil [11].

The largest share of projected land-use change between 2012 and 2030 in both scenarios consists of cropland expansion (excluding sugarcane, see the Supplementary Materials for an overview of land-use changes), which increases by about 320,000 km$^2$ between 2012 and 2030 in both scenarios. This occurs predominantly in the Cerrado biome, at the expense of natural forest, rangeland, and grass/shrubland. Sugarcane expansion also occurs mainly in this region, mostly replacing cropland [6]. Sugarcane area increases by 1400 km$^2$ in the reference situation and 36,775 km$^2$ in the ethanol scenario. In addition, the ethanol scenario projects 20,150 km$^2$ of the indirect land-use change scenario as a result of increased ethanol demand (Supplementary Materials).

### 2.4. Species Habitat Suitability

In this study, biodiversity assessments were based on habitat suitability data for terrestrial mammal species [37]. Habitat suitability data is available for 95% of all terrestrial mammal species at a global scale, covering 5027 species in total [37]. The maps of species habitat suitability used in this study had a resolution of 300 m, and distinguished three habitat suitability classes: Highly suitable, moderately suitable, and not suitable [37]. Only highly suitable habitat, defined as a primary habitat that can sustain the species, was included as a suitable habitat in this study, in line with Visconti et al. [18]. These habitat suitability maps, developed by Rondinini et al. [37] and used as input data in this study, provide habitat suitability per species based on information on habitat suitability from the IUCN Red List [40] on species' range, species minimum and maximum elevation, preferred habitat types (forest, shrub land, grassland, bare land, and artificial), required proximity to water bodies, and tolerance of human disturbance [37], as well as spatial data on land cover (based on Globcover version 2.3 [54]), water bodies, and elevation.

### 2.5. Spatial Analysis of Land Use and Biodiversity

We assessed potential biodiversity impacts of increased ethanol demand using a spatial neighborhood analysis described in an earlier study [36], which combines land-use data with potential species richness data to create projections of biodiversity impact under different scenarios of land-use change. Potential species richness was defined as the sum of all species for which a highly suitable potential habitat occurs in a cell. A map of total potential mammal species richness was created by adding up the habitat suitability maps of all mammal species in Brazil. In a similar way, maps of threatened, endemic, and range-restricted mammal species richness were created by adding up suitability maps of these subsets of species. This method was developed specifically to quantify the impact of land-use change on biodiversity and was applied previously to a different case study [36].

The classification of habitat suitability was based on several factors, including landcover as defined by Globcover v2.3 [54]. To do this, we used the link between habitat suitability and land use to make projections about future potential species richness based on projections of land-use change. To this end, we determined location- and land-use-specific average values of potential species richness, and applied these to the projected land-use map. Because our aim was to create projections of potential species richness based on the PLUC land-use scenarios, the Globcover land cover types were aligned to PLUC land-use types (see the Supplementary Materials for a detailed description of the reclassification). This resulted in the following land-use types being included in the analysis; 'Urban', 'water', 'abandoned land and bare soil', 'forest', 'shrubs', 'grass', 'crops', and 'sugarcane'. The land-use category water contains open water (unsuitable for most terrestrial species) and shorelines. Spatially

variable and land-use specific potential species richness values were then determined by creating a Boolean map of each land-use type, and multiplying these maps with the potential species richness maps, thereby creating a land-use-specific potential species richness map. This provides a map with species richness for each land use, only for the locations where this land use occurs.

A spatial neighborhood analysis was carried out to calculate the potential species richness for each cell in the study area at a 25-km$^2$ resolution. This was done by calculating the average species richness in the cell's neighborhood for a particular land-use type. We used four different window sizes in the calculation: $15 \times 15$ km, $85 \times 85$ km, $395 \times 395$ km, and finally, the whole of Brazil. These window sizes were applied sequentially: If there was a value for the specific land-use type within the smallest window, this window was applied; if not, a larger-sized window was selected (see the Supplementary Materials for a detailed description of the window sizes). For the majority of cells (75%), the $15 \times 15$ km window was applied because a cell with the same land-use type was found within this window size. Potential species richness maps for 2012 and 2030 (both for the ethanol and the reference scenario) were created for all species groups. Potential species richness in 2012 and 2030 for each cell was expressed as the land-use-specific window-averaged number of mammal species with suitable habitat in the direct neighborhood of each cell, from here on referred to as 'species richness index' or SRI. The projected SRI map for 2012 was compared to the 2030 map for the reference scenario to assess changes in SRI due to changes in the demand for agricultural and silvicultural commodities (besides ethanol) during this period. Finally, SRI in 2030 for the reference scenario was compared to SRI in 2030 for the ethanol scenario to assess the impact of ethanol-driven sugarcane expansion on biodiversity.

## 3. Results

### 3.1. Potential Species Richness in 2012

Total potential mammal SRI in 2012 ranged from 0 to 150, depending on the location. In line with an earlier study based on the same potential species range data [37], some hotspots of SRI were identified; SRI was highest in the Amazon region, particularly in the western part (Figure 2), and was slightly lower at the edges of the Amazon rainforest, in the so-called 'arc of deforestation', where forested land mainly borders pasture area. Total SRI was also relatively high in remnants of the Atlantic Forest. SRI was relatively low in the Pampas biome and at the southern edge of the Cerrado region. These areas are dominated by pasture and cropland. Threatened mammal SRI showed a spatial pattern similar to that of total SRI but also had high values in the north-eastern Cerrado region, where more remnant natural vegetation remains. Endemic SRI was highest in the Amazon and eastern Cerrado. Range-restricted SRI was highest along the coastal areas of the Atlantic Forest region, where most forest remnants of Atlantic Forest remain. See the Supplementary Materials for SRI maps for threatened, endemic, and range-restricted species in 2012.

We determined ranges and median values of SRI per land-use type. The median value of the total SRI of forest in Brazil was about three to six times higher compared to the median values for the other land uses (Figure 3). Median total SRI was lowest in sugarcane, urban land, bare land, and cropland. Shrubland, grassland, and water (including shorelines) had slightly higher median values for total SRI. Forest showed the largest variation in total SRI; this can be due to the fact that it includes different forest types, such as Amazonian rainforest and Atlantic Forest, but also plantation forest. Urban area showed the smallest variation. As most land-use types showed a wide range in SRI values, a transition from one land use to another could result either in positive or negative changes, depending on the local SRI values of each land-use type.

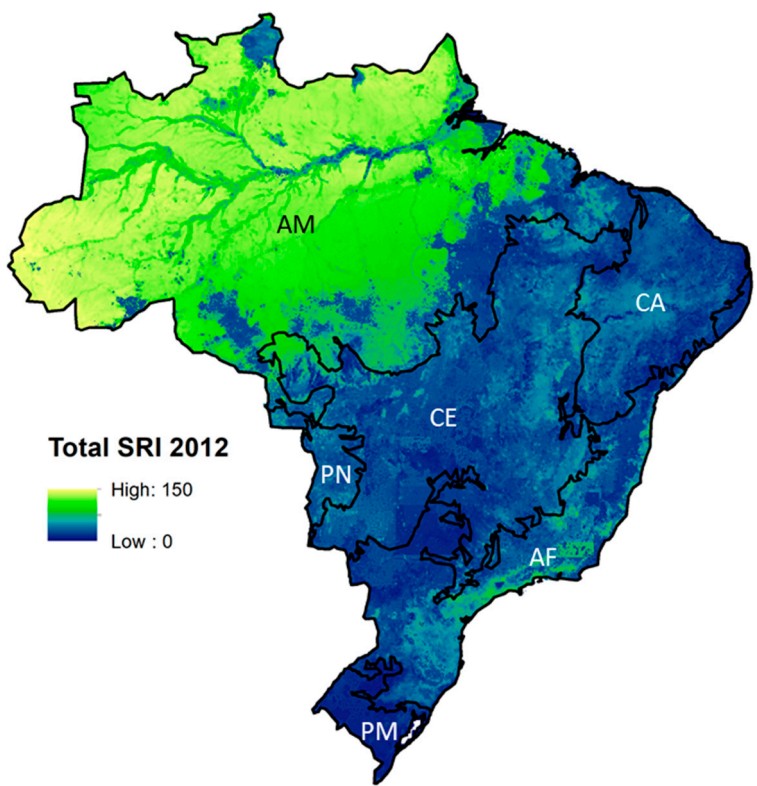

**Figure 2.** Total mammal species richness index (SRI) in Brazil in 2012 at a 25-km$^2$ resolution. Black lines show biome boundaries, abbreviations show biome names; AM = Amazon, CA = Caatinga, CE = Cerrado, PN = Pantanal, AF = Atlantic Forest, PM = Pampas.

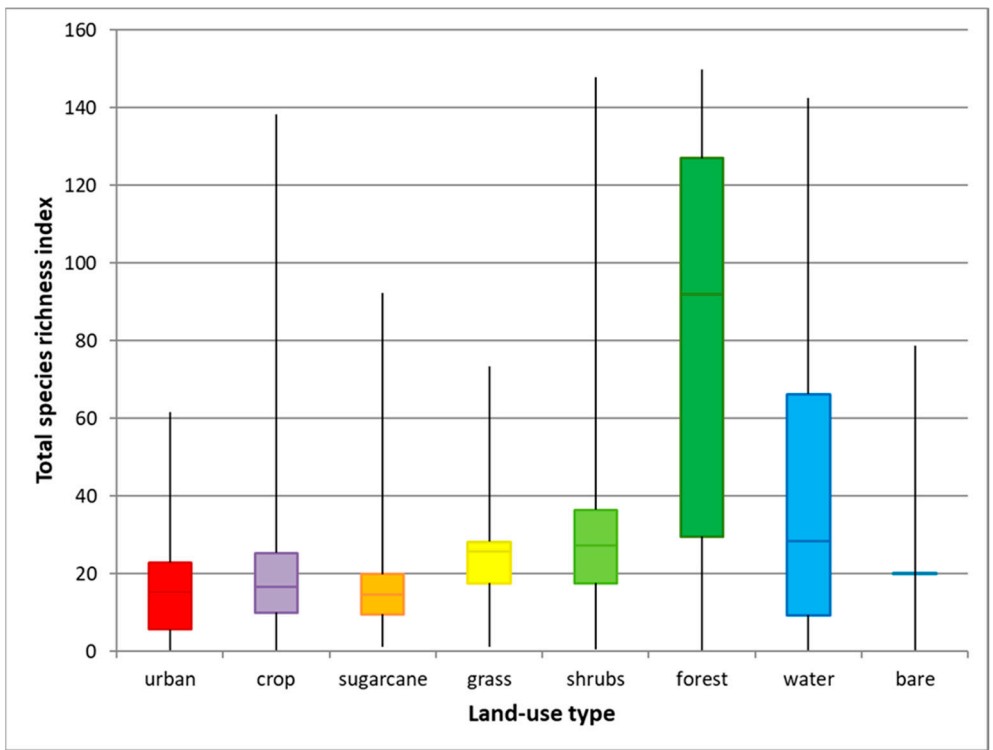

**Figure 3.** Species richness index (SRI) per land-use type in Brazil for 2012. Boxplots show median values (middle horizontal bar) and first (upper limit) and third quartile (lower limit) values, while vertical lines indicate maximum and minimum values. Bare = abandoned land and bare soil.

## 3.2. Projected Changes in Species Richness Index between 2012 and 2030 for the Reference Scenario

Between 2012 and 2030 in the reference scenario, estimated changes in total mammal SRI in Brazil ranged from +115 to −114 (Figure 4). The largest area of SRI changes occurs in the Cerrado biome (Figure 5), including both increases and decreases in SRI. In the Cerrado, reductions in SRI are mainly caused by crop expansion at the expense of grassland, shrubland, abandoned land, and bare soil and forest (see Supplementary Materials for an overview of SRI changes per land-use change), with most changes resulting in a reduction of SRI of over 50% (Figure 5). These areas, however, generally have low SRI values and the absolute changes in SRI are often small (Supplementary Materials). In some areas in the Cerrado, crop expansion results in a small increase in SRI, for example when cropland expands over sugarcane and grassland. In the Amazon region, SRI was reduced mainly due to the projected conversion from forest to grassland (Supplementary Materials). Increases in SRI occur when sugarcane and abandoned land and bare soil are converted into cropland. In the Pantanal and Atlantic Forest biomes, the area with increased total SRI was slightly larger than the area with decreases. The magnitude of positive and negative changes was relatively high in the Amazon biome, where 37% of changes resulted in a loss of total SRI over 50% to 100%. A number of land-use changes, including crop expansion over abandoned land and bare soil and expansion of forest over cropland and grassland, resulted in positive SRI shifts in the Pantanal and Atlantic Forest biomes. For example, forest expansion over cropland resulted in an average increase in total SRI of 8.6 in the Atlantic Forest biome (Supplementary Materials). The intensity (in % change) of changes in SRI were relatively large for range-restricted species compared to the other indicators, with over 20,000 $km^2$ in the Amazon and over 35,000 $km^2$ in the Cerrado projected to lose all range-restricted species. The intensity of changes for endemic and threatened SRI was similar to the magnitude of change for the total SRI but was higher for threatened and endemic species in the Atlantic Forest biome. The intensity of change was higher for range-restricted species.

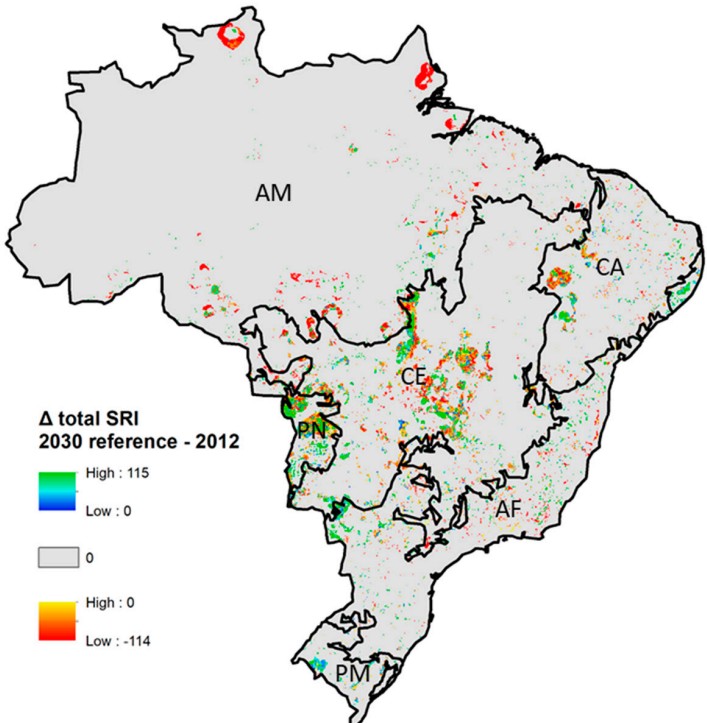

**Figure 4.** Change in total mammal species richness index (SRI) between 2012 and 2030 (reference scenario). Black lines show biome boundaries, biome names: AM = Amazon, CA = Caatinga, CE = Cerrado, PN = Pantanal, AF = Atlantic Forest, PM = Pampas. Grey areas indicate areas with no change in Species Richness Index (SRI).

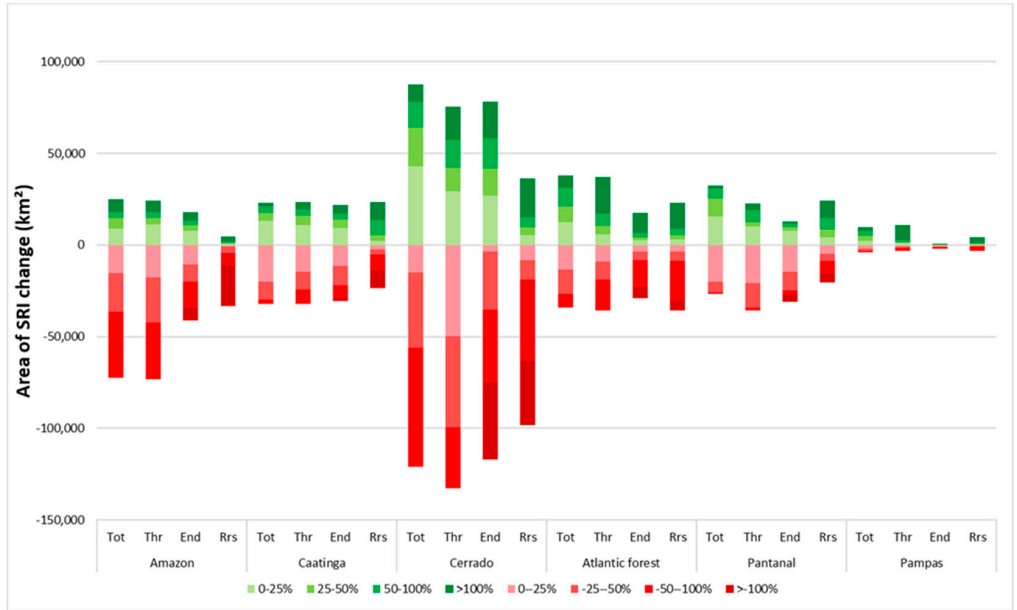

**Figure 5.** Area (in km²) in which changes in the mammal species richness index (SRI) occur between 2012 and 2030 (reference scenario) per biome for total SRI (Tot), threatened SRI (Thr), endemic SRI (End), and range-restricted SRI (Rrs). Decreases in SRI are in red, increases are in green, and darker colors represent a larger percent change in SRI.

*3.3. Projected Differences in Species Richness Index in 2030 between the Ethanol and Reference Scenario*

Differences in SRI between the ethanol scenario and the reference scenario were found predominantly in the Cerrado, Atlantic Forest, and Pantanal biomes, and range from –111 to +103 (Figure 6). The ethanol scenario was projected to have a total SRI 50% lower than the reference scenario in over 2500 km² in the Cerrado, and over 3500 km² in the Atlantic Forest biome (Figure 7). In the Atlantic Forest and Cerrado, lower total SRI in the ethanol scenario was caused by sugarcane replacing forest, grassland, and abandoned land and bare soil. Lower total SRI in the ethanol scenario compared to the reference in the Pantanal occurred where cropland replaces shrubland and forest. Positive shifts also occurred in the Atlantic Forest, which is due to the expansion of forest and grassland, as well as the expansion of sugarcane over cropland. Even though relative differences in total SRI between cropland and sugarcane can be large, absolute SRI differences between crop and sugarcane were small in most regions (Supplementary Materials). A shift from cropland to sugarcane in the ethanol scenario (compared to the reference) resulted in higher SRI in the ethanol scenario in most biomes. In the Pantanal and Amazon biome, however, it resulted in a strong decrease in SRI (Supplementary Materials), because the average total SRI value of cropland in these biomes was about twice as high as in the other biomes (Supplementary Materials). Impacts in the Atlantic Forest and Cerrado region were mostly caused by direct land-use change while impacts in the Amazon, Caatinga, Pantanal, and Pampas were mainly the result of indirect land-use change (see the Supplementary Materials for an overview of SRI changes for direct and indirect land-use change). As a result, changes in biodiversity due to sugarcane expansion should be seen in the context of changes in biodiversity occurring due to other drivers of land-use change. When comparing the ethanol and the reference scenario in 2030, direct land-use change (sugarcane expansion) was responsible for 38% of all land-use transitions, and was responsible for 23% of the decline in total SRI losses (Supplementary Materials). Forest loss (indirect land-use change) constituted 16% of all land-use transitions but was responsible for 27% of all total SRI losses. Between 2012 and 2030, gross sugarcane expansion constitutes about 50,000 km² in the ethanol scenario, compared to about 20,000 km² in the reference scenario (Supplementary Materials). Summarizing, sugarcane expansion was projected to result both in relatively small positive and negative shifts in SRI, depending on the location. Indirect land-use changes, however, consist of

a range of land-use transitions and has a larger impact. Especially, the loss of forest and shrub land resulted in SRI declines.

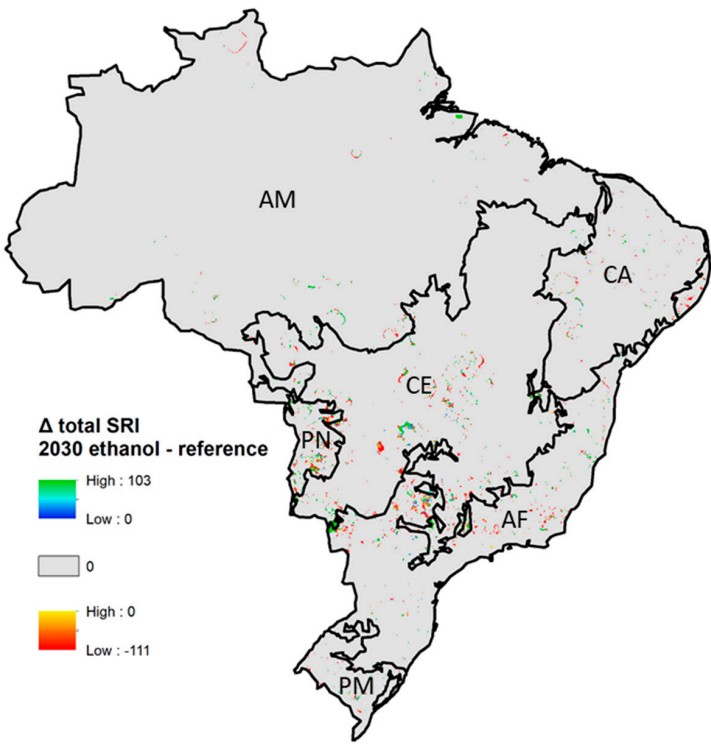

**Figure 6.** Difference in total species richness index (SRI) at a 25-km$^2$ resolution between the ethanol scenario and the reference situation in 2030. Black lines show biome boundaries, abbreviations show biome names; AM = Amazon, CA = Caatinga, CE = Cerrado, PN = Pantanal, AF = Atlantic Forest, PM = Pampas.

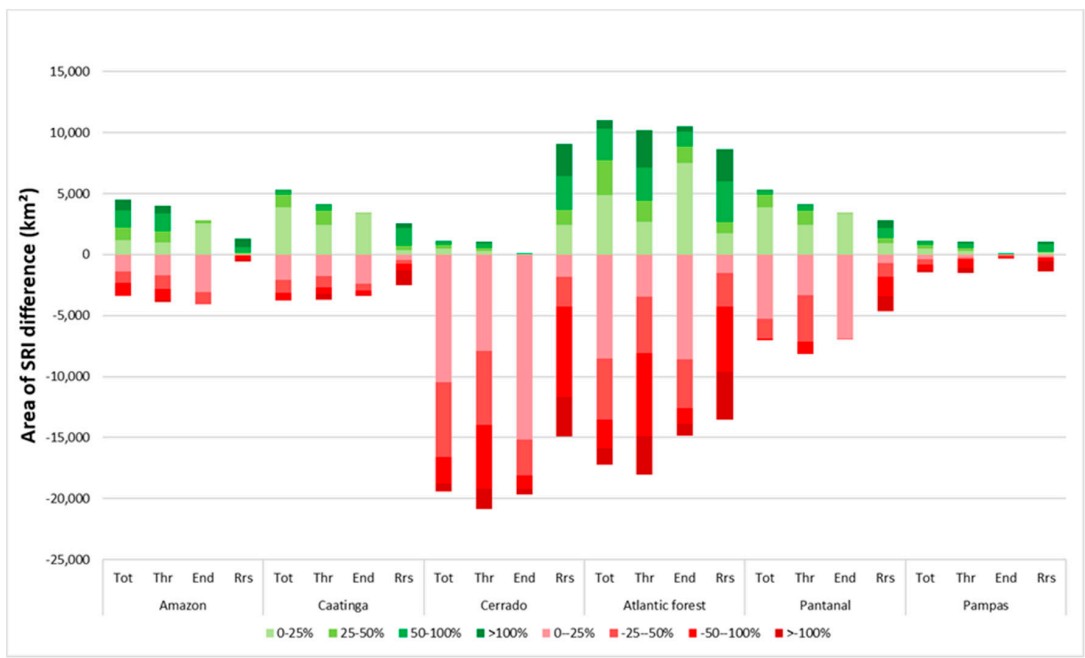

**Figure 7.** Area in which differences in species richness index (SRI) occur in 2030 between the ethanol scenario and the reference, per biome for total SRI (Tot), threatened SRI (Thr), endemic SRI (End), and range-restricted SRI (Rrs). Negative changes are in red, positive changes are in green, and darker colors represent a larger percent change in SRI.

## 4. Discussion

The method applied in this study allows for a spatial and quantitative assessment of biodiversity impacts of land-use change for large groups of species over a large area, and the comparison of different scenarios provides a way to identify impacts due to specific drivers of land-use change. The application of this method to assess biodiversity impacts of increased ethanol demand in Brazil has provided a first indication of potential biodiversity risks and opportunities of ethanol-driven sugarcane expansion. We found that in the absence of increased ethanol demand, Brazil was projected to experience a range of SRI changes due to land-use change caused by increased demand for other agricultural and silvicultural commodities. Biodiversity impacts, both in terms of area and intensity of change, were highest in the Cerrado biome, where cropland is expected to expand, and in the Amazon area, where grassland is expected to replace forest. In the ethanol scenario, expansion of sugarcane was predominantly projected to result in loss of SRI in the Cerrado and Atlantic Forest biomes while crop expansion was expected to result in both positive and negative SRI impacts in the Pantanal region. SRI impacts due to sugarcane expansion were generally small, as it mostly replaces cropland and pasture. Impacts of indirect land-use change showed a large range of potential impacts, highlighting the need to include indirect land-use changes in biodiversity impact assessments. Forest loss led to relatively large losses in SRI, impacts were particularly high when forest was converted to cropland or sugarcane. However, compared to biodiversity impacts due to other land-use changes, biodiversity impacts of sugarcane expansion are relatively small. This implies that other drivers of land-use change need to be addressed as well in order to reduce negative impacts on biodiversity in the future.

Uncertainties in input maps of habitat suitability and land use of Globcover and PLUC influence the results of our analysis. Therefore, our projections of potential species richness should be considered as indicators of trends and areas of interest, rather than robust projections of future biodiversity. Habitat suitability maps contain uncertainty due to the variable extent of knowledge of habitat suitability per region, habitat type, or species [37]. Furthermore, this study did not take into account the landscape composition or connectivity while landscape configuration has been shown to influence local species richness [55,56]. When compared to point locality data for a subset of species, species habitat maps were able to accurately predict about 77% of species occurrences [37]. Validation of the PLUC land-use projections for Brazil has shown that at aggregated spatial levels ($250 \times 250$ km$^2$ and for the whole of Brazil), PLUC has relatively low uncertainty (coefficient of variation, or cv, of 0.91) at allocating direct land-use change, but the precise location of indirect land-use change remains highly uncertain (cv of 1.61) [11]. Projections of land-use transitions towards sugarcane, cropland, and pasture were shown to be relatively accurate in validation of the PLUC model while transitions towards rangeland and planted forest were more uncertain [11].

Our approach also introduced new uncertainty in the analysis. The habitat suitability maps used in this study are, amongst other maps, based on land cover as classified by Globcover 2.3 [54]. This map is based on remote sensing data [57], and its accuracy has been shown to be 67.5%, based on expert validation [54]. Some classes, such as bare soil, cropland, closed broadleaved evergreen forest, and water bodies, were classified more accurately while urban areas, sparse vegetation, and herbaceous vegetation were found to be more prone to misclassification [54]. Aligning land-use classification used in Globcover with the land-use classification in PLUC may have added to the uncertainty. Because the Globcover map contains mosaic land-use types, some assumptions had to be made about the proportion of each PLUC land-use type within these categories.

As shown by our results, SRI values per land-use type show a wide range, and therefore partly overlap. This is due to spatial variation in the potential species richness and the large size of our study area. SRI in cropland differs from one area to the other, for example, due to differences in elevation or proximity to water but also due to heterogeneity within the land-use type due to different crop or management types. For this reason, a specific land-use transition could result in an increase or a decrease in SRI, depending on the location. For example, the replacement of cropland by sugar cane resulted in both positive and negative changes in SRI. This highlights the importance of a spatial

assessment. Future research using this method would be aided by biodiversity data based on more detailed land-use categorization, and more detailed projections of land-use change. We were unable to separate natural grassland from pasture because the Globcover land-use map did not make a distinction between the two types. The availability of maps of managed pasture could reduce the uncertainty of the analysis.

Species richness as an indicator for biodiversity status is simple to calculate and easy to interpret [58], and has been shown to correlate to ecosystem function and resilience [59–63]. However, species richness has also received criticism regarding its ability to reflect the concept of biodiversity and provide information on the conservation priority of areas [64]. An assessment of the number of species does not take into account species composition (beta-diversity) and therefore provides no information on functional diversity, endemism, or rarity [65]. We find that the mean mammal species richness index varies little between most non-forest land-use types. We expect, however, that the species composition varies considerably across these different land-use types. A change from shrubland to cropland may therefore result only in a small reduction of the species richness index while potentially all original shrub-dwelling species are replaced by species tolerant to agricultural lands. This failure to report information on species composition is significant, because community structure and diversity are related to the delivery of ecosystem services [64,66]. In this study, we alleviated this shortcoming by including subsets of species as an indicator, including threatened, endemic, and range-restricted species, which provide some information on the conservation priority of species assemblies. Our study provides projected trends in potential species richness, which require long-term monitoring for verification. We therefore recommend future research to provide field measurements of biodiversity in a range of land-use types, including agricultural lands, such as sugarcane fields.

The focus on potential species richness allowed us to carry out a spatially explicit analysis based on high resolution habitat data [37] for a large number of species. The use of potential habitat data to assess potential species richness avoids problems of sampling effort and observation chances that may skew presence/absence data between different species or regions. However, the presence of suitable habitat is no guarantee of the actual presence of the species that may use this habitat [67]. Activities, such as hunting, may affect the presence of species within suitable habitat [68]. While not explicitly included in the analysis, this aspect was indirectly included in the analysis due to the fact that 'human disturbance' was included as a factor in the habitat suitability maps. The habitat data on which this analysis was based is only available for mammals. It has been shown that different taxonomic groups respond differently to (projected) land-use changes [36,69]. Therefore, results based on species richness for mammals cannot be used as proxy for overall biodiversity.

The results presented here therefore provide first indications of potential areas with high biodiversity loss. However, further research is necessary to assess how other taxonomic groups may be affected by projected land-use changes. As a next step, it will be interesting to apply the method to different mitigation scenarios, such as policy measures that prohibit the transition of forest land, or scenarios that include strategies for nature-inclusive agriculture and agroforestry. Forests were found to have a higher species richness index than the other land-use types. Therefore, in order to reduce the negative impacts of sugarcane expansion on biodiversity, loss of forest should be avoided. Furthermore, to reduce impacts on endemic species, loss of other natural land-use types containing high numbers of endemic species, such as shrubland in the Cerrado biome, should also be prevented. The protection of natural areas can reduce the encroachment of land-use transitions into natural areas, but current protections levels may not be sufficient to avoid impacts; for example, only 2% of the Cerrado biome is legally protected. Biodiversity losses on a landscape scale may furthermore be reduced by efforts to increase biodiversity values of agricultural land-use types. Biodiversity values of cropland could, for instance, be increased by implementing agroforestry practices [70], which could provide suitable habitat for a number of species while also maintaining economically viable yields [71]. More intensive agriculture and livestock production could also reduce the extent of projected indirect land-use changes but also reduce the species richness values of agricultural land. These trade-offs are at the heart of the

land-sparing versus land-sharing debate. More detailed data on species habitat suitability and the spatial distribution of land management would allow for the assessment of these types of mitigation measures. Research has shown that a combination of mitigation strategies (including land conservation policies, yield improvements, and shifts to second generation ethanol production) could reduce the loss of natural vegetation by up to 96% [6]. The quantification of the effect of such mitigation policies and land management scenarios on the biodiversity impact of sugarcane expansion will be the focus of further research.

## 5. Conclusions

This study provides the first quantification of spatial patterns of ethanol-driven biodiversity impacts in Brazil. We applied a method developed specifically to assess spatial trends in biodiversity impacts due to land-use change.

We found that:

- Increased demand for ethanol (from $24 \times 10^9$ L in 2012 to $54 \times 10^9$ L in 2030) resulted in large areas of negative changes in potential species richness in the Cerrado (close to 20,000 km$^2$), Atlantic Forest (over 15,000 km$^2$), and Pantanal (almost 7000 km$^2$) biomes.
- In the Cerrado and Atlantic Forest, about 14% of these changes resulted in losses of total SRI of over 50%. These impacts were mainly due to the direct effects of sugarcane expansion while in the Pantanal, this was due to indirect land-use change.
- These impacts should be viewed in the context of land-use change resulting from increased demand for other agricultural products, which, in the absence of increased ethanol demand, resulted in substantially larger areas with projected biodiversity losses and gains, particularly in the Cerrado (over 200,000 km$^2$, of which over 120,000 km$^2$ negative changes) and the Amazon (close to 100,000 km$^2$, of which over 70,000 km$^2$ negative changes) biomes.
- Loss of forest and shrubland resulted in the largest losses of potential species richness.

Loss of natural vegetation due to sugarcane expansion can be strongly reduced by mitigation strategies, such as reducing the loss of natural vegetation (e.g., through forest protection), increasing biodiversity values of agricultural land (e.g., through agroforestry), or reducing the extent of land-use change (e.g., through agricultural intensification). Future research should focus on assessing potential mitigation options to avoid biodiversity impacts of sugarcane expansion. The identification of locations of potential biodiversity impacts provided in this study is useful for policy makers and ethanol producers aiming to reduce or mitigate biodiversity impacts.

**Supplementary Materials:** The following are available online at http://www.mdpi.com/2073-445X/9/1/12/s1, Figure S1. Area (km$^2$) of projected land use and land-use changes in PLUC projections, Table S1: Sugarcane expansion and other land-use transitions between 2012 and 2030 in the reference scenario, Table S2: Sugarcane expansion and other land-use transitions between 2012 and 2030 in the ethanol scenario, Table S3: Reclassification of Globcover land cover classes to PLUC land use projections, Table S4: Area (km$^2$) and percentage of cells (%) using the window sizes, Figure S2. Window size (in km) applied in the calculation of SRI in 2012 in the spatial analysis, Figure S3. Species richness index in 2012 for threatened, endemic and range-restricted species in Brazil, Figure S4. Species richness index (SRI) per ecoregion and land-use type in Brazil for 2012, Table S5. Summary of the area of difference in land use between the ethanol and the reference scenario and the total SRI losses represented by each category, Table S6. Summary of the area of difference in land use between the ethanol and the reference scenario and the total SRI gains represented by each category, Table S7. Area (in km$^2$) per land-use transition, Table S8. Area (in km$^2$) per land-use transition, Table S9. Species name, reference model number, and their status as threatened, endemic or range-restricted, Figure S5. Difference in species richness index in 2012 and 2030 for the reference scenario, Figure S6. Difference in species richness index in 2030 between the ethanol and the reference scenario, Table S10. Area (in km$^2$) of changes in total SRI in different regions and different impact categories caused by direct (dLUC) and by indirect (iLUC) LUC, and Figure S7. Location of direct (dLUC) and indirect (iLUC) land use changes between the reference and ethanol scenario in 2030.

**Author Contributions:** Formal analysis, A.S.D.; Funding acquisition, A.P.C.F.; Methodology, A.S.D.; Supervision, P.A.V. and F.v.d.H.; Writing—original draft, A.S.D.; Writing—review & editing, A.S.D., P.A.V., A.P.C.F., D.B., C.R. and F.v.d.H. All authors have read and agreed to the published version of the manuscript.

**Funding:** This work was carried out within the BE-Basic R&D Program, which was granted a FES subsidy from the Dutch Ministry of Economic Affairs, Agriculture and Innovation (EL & I).

**Acknowledgments:** The authors would like to thank Maarten Zeylmans van Emmichoven, Yoeri Kraak and Will Zappa for their support in the data analysis.

**Conflicts of Interest:** The authors declare no conflict of interest. The funders had no role in the design of the study; in the collection, analyses, or interpretation of data; in the writing of the manuscript, or in the decision to publish the results.

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
