# Peer review of "Biodiversity Impacts of Increased Ethanol Production in Brazil"

_land, doi:10.3390/land9010012_

Round 1

Reviewer 1 Report

The research work involves a novel approach to the experimentation of biodiversity loss. In addition, the study area is wide enough to include a variety of habitats that enrich the merit of the study. The biological group chosen for the analysis of the models is very attractive both for the specialized public and for the general public, and this aspect is decisive for transferring advances in science to society, and in this case in biodiversity conservation.

Notwithstanding the foregoing, I would like to make some observations and suggestions to the manuscript, which I believe could improve it.

The title seems appropriate, however could it be something more attractive or risky? In addition, the word 'Biodiversity' is repeated in Keywords.

In lines 53, 54, 86, 93, 94, 95, 205, 446, 447, 448 there are references that are not expressed by numbering, is this a publication norm of the journal? To be honest, this format of mixing both styles seems very unattractive. Please unify the references in the text.

Line 138. Can threatened and endemic and endemic species overlap with restricted range? Please explain how the groups of species have been delimited, if they can overlap and depending on the explanation, and if necessary modify table 1.

Line 156 and following. The word Annex does not appear in the supplementary material. Please unify the terminology.

Lines 156, 181, 186 and following. There are a series of double spaces in the manuscript that should be removed so that the body of the text is more aesthetic.

Line 206. Are there no national assessments of threatened fauna that could be integrated into the study? Perhaps, these assessments would change the number of threatened mammal species or the degree of threat.

Line 263. Those Hot Spots have been previously described or are new? That is, do they coincide with other hot spots for the country's biodiversity? If so, please justify this phrase with a reference.

Lines 288 to 296. This whole paragraph is repeated before, so it should be deleted.

Lines 304 to 306, the beginning of this subsection is a repetition of material and methods. It should be deleted or integrates in the right section.

Lines 325, 326. What is the point of differentiation if the intensity of the changes is the same? And what happens in the case of restricted range species? Please clarify this aspect of the results.

Lines 342 and 343, the beginning of this subsection is a repetition of material and methods. It should be deleted or integrates in the right section.

Line 410. Could the uncertainty percentages be listed by region?

Lines 457 to 459. The artifice of including subgroups of species as subrogated from the total group of mammals should be explained in detail in the material and methods section, or in the supplementary material.

The conclusions section is a bit long and repeats some results. I suggest not repeating results or ideas set forth in the discussion section, and perhaps expressing the conclusions in a hierarchical way in the form of bulletpoints or highlighs.

Reviewer 2 Report

In my opinion researches about the impact of the growing area of sugar cane growing on bioethanol and its impact on the number of mammals are very important. On the one hand, sugar cane is the best raw material for the production of the first biofuels generation in economic conditions (OECD-FAO report 2008), which replaces fossil fuels. On the other hand, however, it is important to note and emphasize the ecological effects of constantly growing energy crops. It is important to emphasize the effects of changes in natural vegetation (forests) and replace them by energy crops. Often, these can be irreversible effects.
However, the authors have not avoided minor editorial errors, which I present below.
1. Lines 53, 86, 93, 95 – authors by numbering in accordance with editorial requirements, Adami et al., 2012, Jetz et al., 2007, Faleiro et al., 2013, – are not in the References section.
2. Table 1 horizontal lines
3. Line 446 – Folke, Holling, & Perrings, 1996; Lefcheck et al., 2015; Potter & Woodall, 2014; Soliveres et al., 2016; Zavaleta, Pasari, Hulvey, & Tilman, 2010 – are not in the References section.

Reviewer 3 Report

I would like to congratulate the authors for their work.  I am not in the position to follow all the details of the tools they used, however the overall approach is impressive. The discussion and the conclusions are clearly supported by the results. The only thing that I found that could be improved was the last part of the discussion, lines 477-500. The connection with policy discussions and the overall scientific and social debate on land share /land spare, as well as trade-offs, could be better substantiated. However, this does not affect my final judgment. It is a suggestion for the continuation of their research.

Round 2

Reviewer 1 Report

Abstract

Lines 23 to 28.- In order to unify the text '2' should appear as superscripts

References.

Please justify the text

Lines 529, 532, 539, 560, 562, 621, 648, 656, 687. Please finish the reference with a final point.

Supplementary

Lines 129 and.147- 'Table A8', 'A6'. The font color is red

Author Response

Lines 23 to 28.- We have adapted '2' to appear as superscripts

References - we have justified the text

Lines 529, 532, 539, 560, 562, 621, 648, 656, 687. We have finished the reference with a final point.

Lines 129 and.147- 'Table A8', 'A6'. The font color is changed to black